# Simulation of 69 microbial communities indicates sequencing depth and false positives are major drivers of bias in prokaryotic metagenome-assembled genome recovery

**Ulisses Rocha**[1‡*], **Jonas Coelho Kasmanas**[1‡], **Rodolfo Toscan**[1], **Danilo S. Sanches**[2], **Stefania Magnusdottir**[1], **Joao Pedro Saraiva**[1*]

**1** Department of Applied Microbial Ecology, Helmholtz Center for Environmental Research-UFZ, Leipzig, Germany, **2** Department of Computer Science, Federal University of Technology—Paraná, UTFPR, Cornélio Procópio, Brazil

‡ These authors share first authorship on this work.
* ulisses.rocha@ufz.de (UR); joao.saraiva@ufz.de (JPS)

**Data Availability Statement:** The true positive metagenome-assembled genomes (MAGs) (4638

## Abstract

We hypothesize that sample species abundance, sequencing depth, and taxonomic relatedness influence the recovery of metagenome-assembled genomes (MAGs). To test this hypothesis, we assessed MAG recovery in three in silico microbial communities composed of 42 species with the same richness but different sample species abundance, sequencing depth, and taxonomic distribution profiles using three different pipelines for MAG recovery. The pipeline developed by Parks and colleagues (8K) generated the highest number of MAGs and the lowest number of true positives per community profile. The pipeline by Karst and colleagues (DT) showed the most accurate results (~ 92%), outperforming the 8K and Multi-Metagenome pipeline (MM) developed by Albertsen and collaborators. Sequencing depth influenced the accurate recovery of genomes when using the 8K and MM, even with contrasting patterns: the MM pipeline recovered more MAGs found in the original communities when employing sequencing depths up to 60 million reads, while the 8K recovered more true positives in communities sequenced above 60 million reads. DT showed the best species recovery from the same genus, even though close-related species have a low recovery rate in all pipelines. Our results highlight that more bins do not translate to the actual community composition and that sequencing depth plays a role in MAG recovery and increased community resolution. Even low MAG recovery error rates can significantly impact biological inferences. Our data indicates that the scientific community should curate their findings from MAG recovery, especially when asserting novel species or metabolic traits.

out of 5217) obtained in this study are available at the National Center for Biotechnology Information (NCBI) repository with the sample accession numbers SAMN34004744 - SAMN34012094 (BioProject PRJNA950613). The complete set of MAGs (including the false positives) and simulated metagenomic libraries are available on the long-term data archive at the Helmholtz Center for Environmental Research – UFZ data center using the link (https://www.ufz.de/record/dmp/archive/13476). All the code used during data analysis is available at https://github.com/mdsufz/mockc_analysis

**Funding:** This work was funded by the Helmholtz Young Investigator grant VH-NG-1248 Micro' Big Data' and the Deutsche Forschungsgemeinschaft (DFG, German Research Foundation) – project number 460129525, and these grants were awarded to UR. JCK was supported by São Paulo Research Foundation (FAPESP; grant 2019/03396-9 and 2022/03534-5). The funders had no role in study design, data collection and analysis, decision to publish, or preparation of the manuscript.

**Competing interests:** The authors have declared that no competing interests exist.

## Author summary

Microbial communities are incredibly diverse and play essential roles in ecosystems, from recycling nutrients to influencing climate change. We explored how the microbial community assembly can influence its species' metagenomics recovery. Specifically, we examined how the abundance of different species within a sample, the extent of DNA sequencing (sequencing depth), and the species taxonomic relatedness affect our ability to accurately reconstruct these communities.

We computationally simulated three microbial communities, each composed of 42 species. These communities varied in species abundance, sequencing depth, and how closely related the species were to each other. We then applied three different computational techniques to reconstruct the original communities from the simulated sequence data.

Our findings highlight the critical impact of sequencing depth and taxonomical relatedness, specifically, on accurately recovering microbial genomes. Interestingly, more sequencing does not always equate to more accurate community representation. Moreover, even a few false positives can significantly distort our interpretations of microbial diversity and function.

Our research underscores the importance of carefully considering these factors in metagenomic studies to avoid misleading conclusions about microbial ecosystems. Our work contributes to refining metagenomic techniques, aiming for a more reliable and nuanced understanding of microbial life's role in our planet's health and functioning.

## Introduction

Microbial communities contribute to ecosystems by performing a wide array of ecosystem processes such as the decomposition of organic matte [1], degradation of contaminants [2] or carbon and nitrogen cycling [3,4]. The use of metagenomics is a standard method to study microbial ecology, human health, and environmental issues, with several databases emerging to facilitate data selection depending on the research questions [5–7]. This use is shown by the exponential increase in deposited Metagenome-Assembled Genomes (MAGs) in public repositories. For example, the National Center for Biotechnology Information (NCBI) started with 132 in 1993 but had over 1.6 million genomes deposited when we wrote this article [8]. The Joint Genome Institute (JGI) started with 35 complete or draft genomes in 1997 but currently has 411829 [9]. Advances in Next Generation Sequencing (NGS) and the development of bioinformatics pipelines allow the recovery of near-complete genomes from complex samples, which can be used to characterize microbial communities and assert their functional potential [10,11]. For example, Parks and colleagues [12] recovered nearly 8000 metagenome-assembled genomes (MAGs) using MetaBAT [13]. Genome annotation of such a large number of MAGs allows us to infer the functional potential of species across a diverse array of environmental conditions and provides a wide range of applications. For example, over 12000 studies performed in 2022 rely on MAGs [14]. The assertions of microbial community composition and their functional profile are thus tightly linked to the reliability of the tools and methods employed to generate MAGs [15]. Systematic errors during MAG recovery may profoundly impact the assertions made in microbial community studies that use metagenomics [16]. Previous studies have evaluated the effect of individual parameters on genome recovery. Gweon and colleagues [17] showed that sequencing depth affected the profiling of microbial communities, while Anyansi and colleagues [18] highlighted the challenges in recovering closely

related species. However, studies that simultaneously tackle the different levels of complexity in microbial communities, such as species abundance and species diversity, are yet to be performed. Minimizing the potential systematic errors during MAG recovery will benefit their growing cascade of ecological and biotechnological applications. With the massive increase in data generated from MAG analyses, several questions remain unanswered; among them: (i) how reliable are the genomes recovered from metagenomes?; (ii) how much of the microbial communities are we missing?; (iii) do the presence of Eukaryote genomes creates bias in the recovery of prokaryotic MAGs?; and (iv) how do sequencing depth, species abundance distribution, and taxonomic relatedness between species influence genome recovery?

Reconstruction of MAGs from all domains of life (prokaryotic, eukaryotic, and viruses) is mostly performed separately. Pipelines such as those employed by Parks and colleagues [12], Sieber and co-workers [19], and Albertsen and colleagues [20] are just a few examples of prokaryotic MAG recovery. Recent pipelines have been developed for the recovery of eukaryotes (EukRep [21]) and viruses (VirSorter [22] and VIBRANT [23]). To our knowledge, only one framework exists that recovers genomes from metagenomes from all domains of life [24]. Independently of the domain of life, much more work needs to be done to increase confidence in the characterization of microbial communities and determine how much of the communities are left uncharacterized. Community diversity and sequencing depth have been suggested to play essential roles in MAG recovery [16]. Quality assessment of Prokaryotic MAGs usually relies on sets of single-copy genes (SCGs) or reference genomes to generate results [25]. Using only one single-copy gene set may limit the pipelines' ability to capture all species in a microbial community. For example, strain heterogeneity [26] and uneven population abundances of organisms within a community have influenced MAG recovery. The Critical Assessment of Metagenome Interpretation (CAMI) [27] challenges attempts to create benchmarks for comparing metagenomic tools since they are usually difficult to compare. Simulated communities described by Meyer and co-workers [27] mimic some properties of microbial communities, such as multiple closely related strains or species and abundance profiles. Nevertheless, to our knowledge, no dataset exists that contemplates a combination of taxonomic relatedness, species abundance, sequencing depth, and the use of species with multiple or linear chromosomes.

In this study, we tackle MAG recovery knowledge gaps by generating mock communities with varying parameters of sequencing depth, taxonomic relatedness, and species abundance profiles. We use these datasets to evaluate three pipelines for binning, each using a different approach to bin the sequences, and assess which factors or combinations of factors drive Prokaryotic MAG recovery and if these are pipeline-dependent. We also evaluate the effect of multiple or linear chromosomes in MAG recovery.

## Results

### Short description, number of MAGs, quality assessment of simulated communities, and 16S rRNA recovery

The definitions of species abundance distributions, taxonomic relatedness, and sequencing depth used in this study are detailed in the methods section. In short, the study created 69 simulated microbial communities divided into three communities (A, B, and C). A total of 126 species were selected and divided into three communities of 42 (33 bacteria, six archaea, and three fungi) (S35 Table). Species abundance distribution is varied as: equal, logarithmic decay, exponential decay, logarithmic decay with abundance plateaus, and exponential decay with abundance plateaus. Taxonomic relatedness is classified as "Very closely related" (same genus), "Closely related" (same family, different genus), or "Not closely related" (S30 Table).

Sequencing depth is explored at five levels: 10, 30, 60, 120, and 180 million paired-end Illumina reads. Communities are constructed as "Ordered," "Random," "Very closely related," "Closely related," and "Not closely related". We define a "community profile" as a given combination of the mentioned parameters (e.g., community A, exponential abundance decay distribution, very closely taxonomical relatedness, 60 million paired-end Illumina reads) used to simulate a community. Finally, we evaluated 3 MAGs recovery pipelines: the multi-metagenome pipeline, hereafter referred to as MM [20]; the DAS Tool pipeline, hereafter referred to as DT [19]; and the pipeline developed by Parks and co-workers, hereafter referred as 8K [12].

**Number of MAGs and quality assessment.**   The pipelines recovered a total of 2915 (8K), 1429 (DT), and 873 (MM) MAGs from the three communities (S1 Table). Considering all community profiles, the 8K pipeline recovered, on average, the highest number of MAGs (42.24, ±122.35), with MM recovering the lowest (12.65, ±27.07). As expected, no eukaryotic genomes were recovered. Average MAG completeness was 96.74 (±4.99) (8K), 97.56 (±4.45) (DT) and 97.32 (±3.49) (MM), and average contamination was 0.363 (±0.505) (8K), 1.25 (±1.65) (DT) and 1.10 (±1.61) (MM) (S1 Fig and S1 Table). On average, the DT and MM pipelines only recovered a single MAG per species, while the 8K recovered 3.59 (±0.29) (S2 Table).

**16S rRNA recovery.**   The 8K pipeline produced 10 gene sequences, of which 6 were unique rRNA gene sequences in the MAGs. The MAGs produced by the DT pipeline yielded 1393 rRNA gene sequences, of which 123 were unique. Furthermore, 900 rRNA gene sequences were assigned to bacteria, 73 to archaea, and 32 to eukaryotes (S3 Table). Sankey plots with the species found using the 8K, DT, and MM pipelines in all communities (A, B, and C) (S4 Table) are shown in (S2–S10 Figs).

## Impact of sequencing depth on the number of MAG recovery

From a sequencing depth of 10 million to 60 million reads, an increasing trend in the number of MAGs recovered was observed across all three pipelines, irrespective of the community profile (Fig 1A and 1D). The 8K pipeline recovered the most MAGs in all sequencing depths, followed by the DT and the MM pipelines (S12–S14 Tables). However, increasing the sequencing depth further did not consistently increase the MAG recovery.

The sequencing depth of 60 million reads appears to be the optimal point for MAG recovery. Beyond this depth, the MM pipeline showed fewer MAGs recovered at 120 and 180 million reads (S11 Fig and S8 Table). For the DT pipeline, sequencing beyond 60 million reads did not yield a significant increase in MAG recovery, indicating a plateau effect at this depth (S13 Fig and S10 Table). Finally, although the 8K pipeline continued to recover significantly more MAGs than the other pipelines (bonferroni adjusted t-test, $p < 0.05$), it also did not increase its absolute number recovered after 60 million reads (Fig 1A and 1D).

## Influence of taxonomic relatedness on MAG recovery

The definition of taxonomic relatedness used in this study is detailed in the methods section. Briefly, taxonomic classification was used to create "species pairs" (S30 Table). A species was randomly selected for a given phyla to serve as the reference. Then, species were grouped into triplets based on their taxonomic proximity to this reference. Species of the same genus as the reference were classified as 'Very closely related.' Species that shared the same family but not the genus with the reference were categorized as 'Closely related.'

In 'Closely related' communities, the 8K, DT, and MM pipelines showed the respective average' species pairs' recovery rates: 0.11% (±0); 0.31% (±0.09); 0.20% (±0.048) (Fig 2 and S29 Table). Notably, following an exponential decay species abundance distribution, the highest recovery rates for 'species pairs' in these 'closely related' communities were observed.

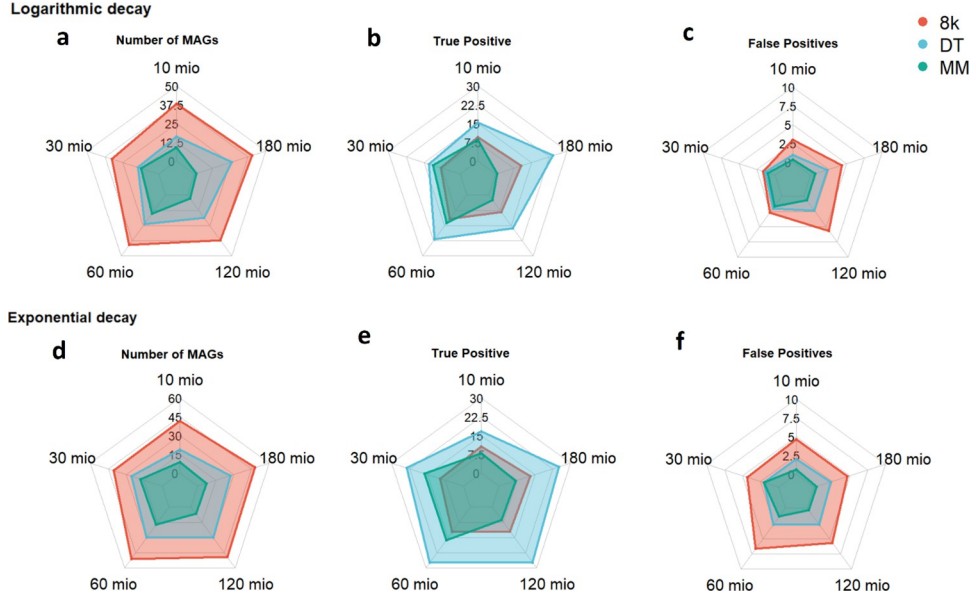

**Fig 1.** Radar plots for the Number of recovered MAGs (a, d), number of recovered MAGs that were present in the original community (b, e)–or True Positives MAGs–and number of recovered MAGs not present in the original community (c, f)–or False Positives–for a given sequencing depth (e.g. 60 mio reads) for the analyzed pipelines (8K, DT, and MM). Radar plots a-c are from communities with logarithmic decay species abundance profile, and plots d-f obtained from the Exponential decay profile. Taxonomic relatedness was kept as Random.

In 'Very closely related' communities, the 8K pipeline recovered an average of 0.076% (±0.04) of 'species pairs' (S29 Table). The DT pipeline recovered 0.26% (±0.01) of the, and the MM pipeline recovered 0.26% (±0.041) (Fig 2 and S29 Table). Furthermore, in these 'very closely related' communities, both the DT and MM pipelines significantly outperformed the

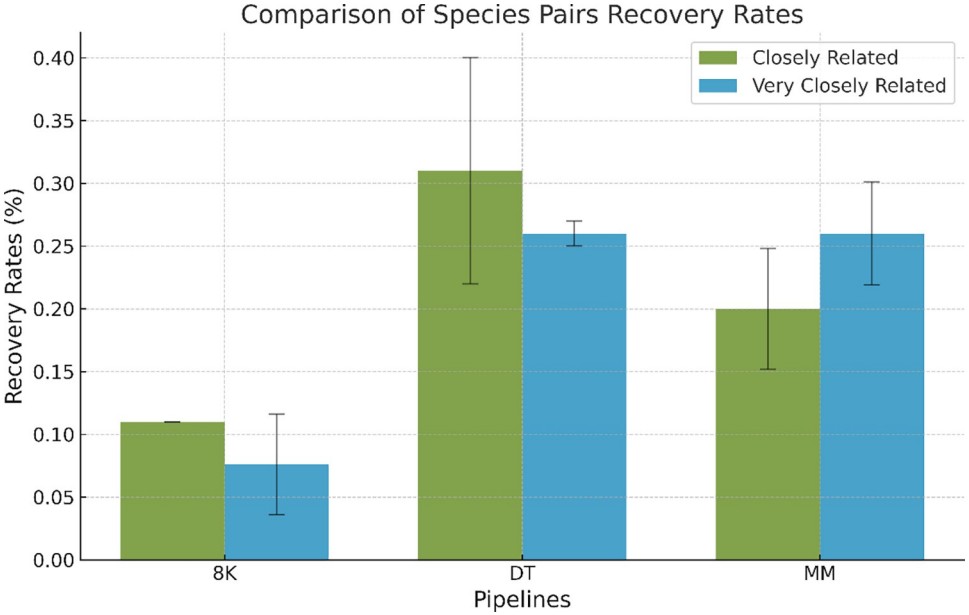

**Fig 2. Histogram comparing species pair's recovery rates in Closely and Very Closely Related communities across different pipelines (8K, DT, and MM) regardless of the community profile.**

8K pipeline in recovering 'species pairs', particularly when the species abundance distribution followed a logarithmic decay (bonferroni adjusted t-test, $p < 0.05$) (S20 Fig and S34 Table).

Complete results for the recovered 'species pairs' across different pipelines and community types are compiled in S31–S33 Tables for the 8K, DT, and MM pipelines, respectively. This data underscores the nuanced relationship between taxonomic proximity and the efficiency of different pipelines in recovering similar taxa.

## Effects of species abundance profile on MAG recovery

The definition of the species abundance profile used in this study is detailed in the methods section. Briefly, species abundance refers to each species' relative amount of genomic data within a community. The species abundance profile in genomic data can be evenly, normally, or disruptively distributed. We simulated various states of species abundance distribution, including equal species abundance, logarithmic decay, exponential decay, and both logarithmic and exponential decay with abundance plateaus (S21 Fig).

For the logarithmic decay profile, the DT pipeline significantly outperformed the MM pipeline in recovering MAGs in communities sequenced at 10, 60, and 180 million reads (Fig 1A and S13 Table). Additionally, in communities characterized by a logarithmic decay with abundance plateaus, particularly those not closely related, the DT pipeline also recovered more MAGs than the MM pipeline (Fig 3 and S14 Table).

Communities with an exponential decay in species abundance distribution consistently yielded more MAGs. This trend was particularly pronounced in both 'Very closely related' and 'Closely related' communities. Compared to the logarithmic decay profile, the exponential decay profile resulted in a significantly greater number of recovered MAGs for 'Very closely related' communities (S12 Fig and S9 Table) and for 'Closely related' communities (S14 Fig and S11 Table).

Finally, the 8K pipeline showed no significant MAG recovery differences across species abundance profiles. Whether considering taxonomic relatedness (S5 and S6 Tables) or sequencing depth (S7 Table), the 8K pipeline's performance remained consistent across all community profiles.

## Analysis of true and false positive effects

In assessing the binning performance of each pipeline in various community profiles, we calculated true positive (TP), false positive (FP), and false negative (FN) metrics, with detailed methodologies described in the methods section. A TP is a MAG correctly recovered from the original community, an FP is a recovered MAG not present in the original community, and an FN is an unrecovered MAG in the original community. A detailed description of the performance assessment can be found in the methods section.

**True positive recovery.** On average, the 8K pipeline recovered 25% of the 42 original species in each community profile (S2 Table). When examining 'Closely related' communities, this pipeline had a significantly (bonferroni adjusted t-test, $p < 0.05$) lower number of TPs in communities with a logarithmic decay species abundance distribution compared to those with exponential decay and abundance plateaus (S15 Table). Notably, no significant differences in TP recovery were observed when comparing sequencing depths or taxonomic distributions within any species abundance distribution profile (S16 and S17 Tables).

The DT pipeline emerged as the most effective, recovering, on average, 45% of the original species, significantly higher than the other pipelines (S2 Table). Moreover, it recovered 85% of the original species at least once across the community profiles (S18 Table). In 'Closely related' communities, this pipeline obtained a higher number of TPs in communities following an

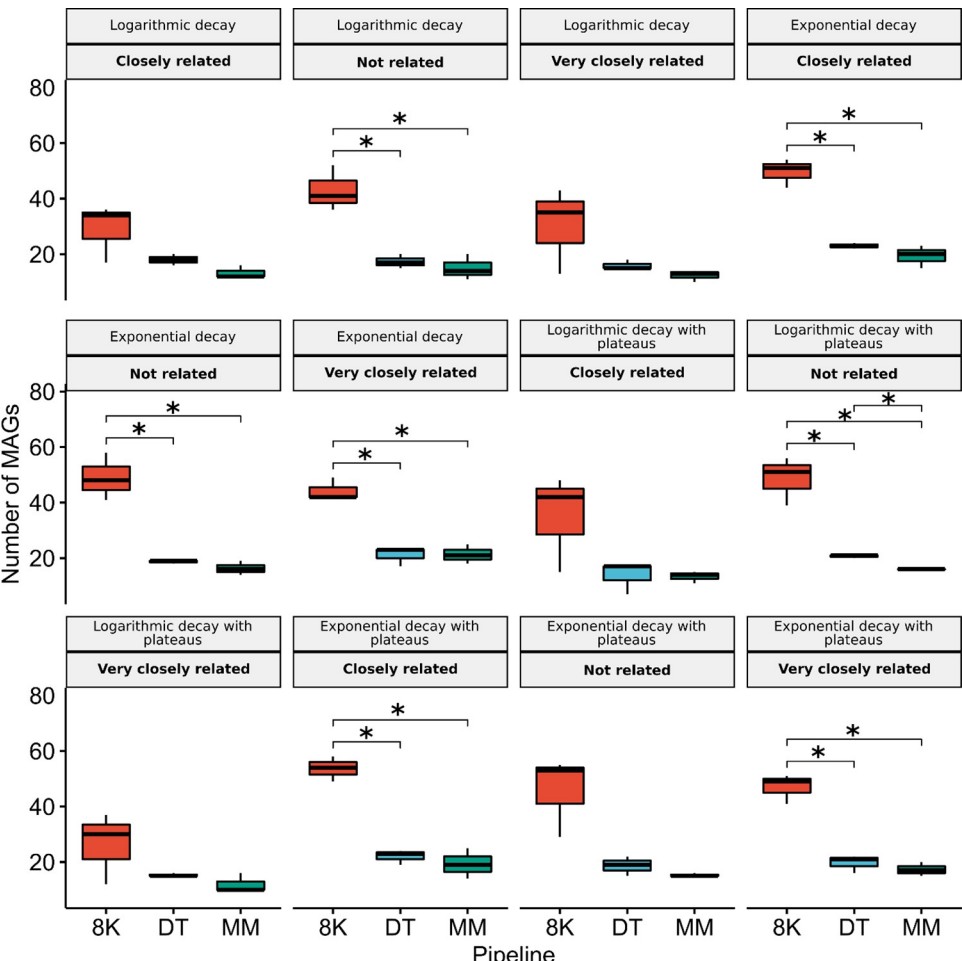

**Fig 3. Student's Bonferroni adjusted t-test comparing metagenome-assemble genome counts between all pipelines used (8K, DT, and MM) according to Taxonomic relatedness (Not related; Closely related; Very closely related) and species abundance distribution (Logarithmic decay; Exponential decay; Logarithmic decay with abundance plateaus; Exponential decay with abundance plateaus).** The sequencing depth of the communities was kept at 60 million reads. (*P-value < 0.05).

exponential decay compared to those with a logarithmic decay. However, a lower TP count was observed in communities sequenced at the lowest depth of 10 million reads (S15 Fig), and no significant increase in TP was observed after 60 million reads (Fig 1B and 1E, and S19 and S20 Tables).

The MM pipeline recovered, on average, 27.5% of the original species (S2 Table). In "very closely related" communities, a significantly higher number of TPs were observed under exponential decay species abundance distributions when compared to those with logarithmic decay (bonferroni adjusted t-test, p < 0.05) (S16 Fig and S21 Table). It achieved an increasing number of TPs at sequencing depths of up to 60 million reads, and this number decreased significantly in communities sequenced at 120 and 180 million reads (Figs 1B, 1E and S17, and S22 Table).

Comparatively, the DT pipeline outperformed both the MM and 8K pipelines in recovering the original species (Fig 1B and 1E), irrespective of sequencing depth, taxonomic relatedness, or species abundance distribution (Figs 4, S18 and S22, and S24–S26 Tables). Additionally, the 8K pipeline recovered fewer TPs than the MM pipeline in communities with an exponential

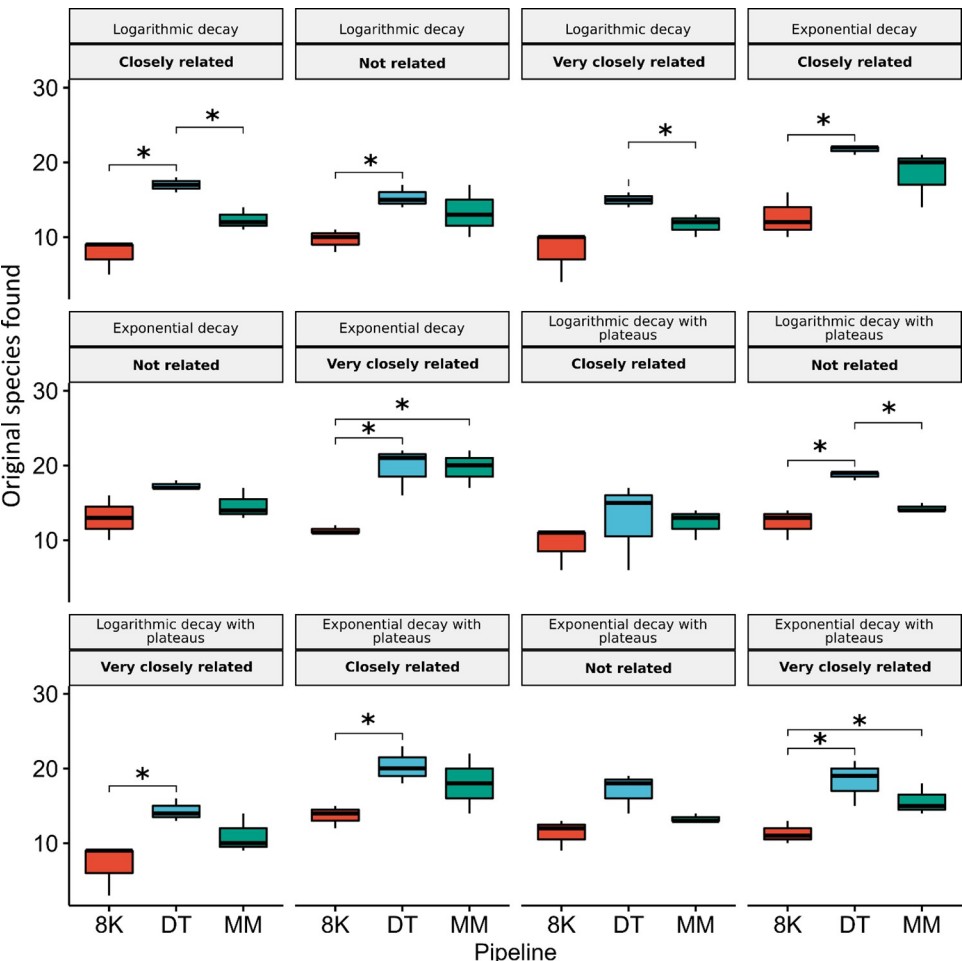

**Fig 4. Student's Bonferroni adjusted t-test comparing True Positives Species recovered and in the original communities between all pipelines used (8K, DT and MM) according to Taxonomic distribution (Not related; Closely related; Very closely related) and species abundance distribution (Logarithmic decay; Exponential decay; Logarithmic decay with abundance plateaus; Exponential decay with abundance plateaus).** The sequencing depth of the communities was kept at 60 million reads. (*P-value < 0.05).

decay at 30 and 60 million reads. In comparison, the MM pipeline recovered fewer TPs than the 8K pipeline at higher reads of 120 and 180 million (S18 Fig and S26 Table).

**False positive recovery.** The 8K, DT, and MM pipelines yielded FP rates of 8.8%, 8.1%, and 9.4%, respectively, of the 42 species in the original communities (S2 Table). Notably, the 8K pipeline produced significantly more false positives than the DT pipeline in 10 out of 23 community profiles and more than the MM pipeline in 7 profiles. These significant differences were observed in communities sequenced at 60 million reads (Fig 1C and 1F). No clear pattern was observed when analyzing taxonomic relatedness and species abundance distribution, suggesting that both parameters do not significantly influence the number of false positives.

The MM pipeline demonstrated the lowest number of false positives, with only 75 FPs, compared to 280 for the 8K pipeline and 117 for the DT pipeline. This trend was consistent regardless of the species abundance distribution and sequencing depth (S19 Fig and S27 Table). When comparing the number of unique species identified as false positives, the 8K, DT, and MM pipelines identified six, nine, and eight unique species, respectively. Further-more, all three pipelines consistently misidentified five specific species–*Bifidobacterium*

*vaginale*, *Bordetella pertussis*, *Porphyrobacter atlanticus*, *Pseudothermotoga caldifontis*, and *Rhizobium rhizogenes*–as false positives. Regarding species-specific recovery, *Desulfurococcus amylolyticus* was falsely identified by both the DT and 8K pipelines. At the same time, the DT and MM pipelines falsely recovered *Borreliella burgdorferi*, *Caulobacter vibrioides*, and *Trichormus azollae* (S4 Table).

We did not identify contigs in the Prokaryotic MAGs from the Eukaryotic genomes added to the simulated microbial communities.

**Accuracy recovery rate.** The DT pipeline displayed the highest average accuracy in species recovery (~92% ±2.07), followed by the MM pipeline (~88% ±15.57). The 8K pipeline displayed the least accuracy in species recovery (~76% ±8.81) (S2 Table). The DT pipeline was significantly more accurate than the 8K pipeline in 10 of the 23 community profiles. In comparison, the MM pipeline showed significantly higher accuracy rates than the 8K pipeline in 6 of the 23 community profiles (bonferroni adjusted t-test, $p < 0.05$). The most significant differences in both cases (i.e., DT versus 8K and MM versus 8K) were observed in communities sequenced at 60 million reads. The results did not show a clear pattern in accuracy when analyzing species abundance distribution and taxonomic relatedness.

## Detection limit

This study investigated the abundance detection limits for MAG recovery across the different community profiles using different pipelines. Detailed information on the generated species abundances for all original community profiles can be found in the methods section (S37 Table). Briefly, the lowest abundant generated sequences have 0.009 relative abundance. The abundance increase follows the abundance distribution for the given community profile.

Using the 8K pipeline, notably, *Ferroglobus placidus*, an archaea, was detected with the lowest TP relative abundance (0.0003) and second-lowest genome coverage (4) in a 'Random' community with logarithmic decay, sequenced at 180 million reads. In the same community profile, *Desulfurococcus amylolyticus*, another archaea, was erroneously recovered (FP) with a higher relative abundance (0.022) and coverage (549). For the DT pipeline, the bacterium *Wolinella succinogenes* showed the lowest TP relative abundance (0.00046) and coverage (3) in 'Very closely related' communities following a logarithmic decay, sequenced at 60 million reads. Conversely, *Caulobacter vibrioides*, an FP, was recovered in a similar community with a higher relative abundance (0.0821) and coverage (216). Intriguingly, in 'Random' communities with logarithmic decay sequenced at 180 million reads, *C. vibriodes* exhibited the highest false positive coverage (1199) and relative abundance (0.1536). Finally, utilizing the MM pipeline, the *Borrelia anserina* had the lowest TP relative abundance (0.00073) and the third-lowest coverage (10x) in 'Very closely related' communities following logarithmic decay, sequenced at 60 million reads. *Granulicella mallensis* presented the lowest TP coverage (8x) among the original species in 'Random' communities with exponential decay, sequenced at 10 million reads. *C. vibriodes*, not included in the original species list (FP), showed a high relative abundance (0.148) and coverage (406x) in 'Closely related' communities with logarithmic decay, sequenced at 60 million reads.

The relative abundances of all original and recovered species across each community profile and using all pipelines are comprehensively detailed in S4 Table.

## Special genome structures and recovery challenges

The community creation and definition are described in the methods section. These community definitions and relationships did not include fungi and special cases of bacteria genome topology (consisting of linear or multiple chromosomes). However, representatives of species

with linear (*Vibrio cholerae* and *Rhodobacter sphaeroides*) and multiple chromosomes (*Bacillus subtilis*, *Streptomyces griseus*, *Borrelia burgdorferi*, *Paracoccus denitrificans*, *Planktomarina temperate*, *Vibrio fluvialis*, and *Brucella melitensis*) were included in each simulated community. The list of all species used per community and type of chromosome is available in S35 Table. Following, we analyzed the recovery of these 'special cases'.

As expected, none of the 'special case' genomes (i.e., linear or multi-chromosomes) belonging to fungi were recovered. All bacterial "special cases" were recovered at least once by each pipeline except *Borrelia burgdorferi* and *Brucella ovis* (S28 Table). The MM pipeline could not recover any 'special case' genomes in communities composed of species at equal abundances (S4 Table). Interestingly, *Brucella melitensis* was recovered in the original communities and others irrespective of the pipeline used, the species abundance, and taxonomic distribution or sequencing depth (S4 Table). On average, 62.5% (±13.5) of the bacterial 'special cases' were recovered across all community profiles (S4 Table).

## Discussion

This study aimed to determine which factors influenced genome recovery from metagenomes, the detection limits of MAG recovery, and which of the three binning tools performed best in MAG recovery. Additionally, we included fungi in our simulated microbial communities to determine their impact on the recovery of prokaryotic MAGs. Our prokaryotic MAGs showed no contamination from the genetic content of the eukaryotic genomes, indicating the presence of eukaryotes does not impact the recovery of prokaryotic MAGs even though the presence of eukaryotes was confirmed though the reconstruction of eukaryotic rRNA sequences. Our results showed that MAG recovery was influenced by taxonomic relatedness and sequencing depth but not species abundance. For example, more MAGs were recovered by the DT pipeline in communities composed of closely related species than non-related species (following an exponential decay species abundance distribution) than by the 8K and MM pipelines. Sequencing depth did appear to influence MAG recovery in the DT and MM pipelines, but no clear pattern was observed. For example, the higher number of MAGs with increasing sequencing depths using the DT pipeline was not observed when using the MM pipeline, which depended on the combination of species abundance distribution and sequencing depth. The MM pipeline uses differential binning coverage [20] (i.e., the abundance of species in the samples), which would lead to higher binning resolution with increased sequencing depths. However, our results aligned with studies evaluating sequencing depth [28,29], which suggested that higher sequencing depths do not necessarily translate to improved species recovery. Our results suggest a sequencing depth sweet spot of 60 million reads. Our tool comparison showed that the 8K pipeline always yielded more MAGs than the DT and MM pipelines, which may be linked to the use MetaBAT [13]. MetaBAT combines tetra-nucleotide frequency and contig abundance probabilities during the binning process. However, it also recovered a high number of genomes not originally present in the community (False positives).

Another aspect of this study was to examine how much of the original species (True Positives) added to the simulated communities each pipeline could recover. Sequencing depth showed contrasting results between the DT and MM pipelines. Higher sequencing depths in the DT pipeline resulted in increased TPs, but without significant increase after 60 million reads. The MM pipeline showed increased TP recovery in sequencing depths up to 60 million reads compared to 120 and 180 million reads. This result suggested that the combination of multiple binning tools used by the DT pipeline scales with sequencing depth, contradicting the expected higher binning resolution with increased sequencing depths. Besides, using tools that rely on species abundances showed that higher sequencing depths did not result in the

recovery of a higher number of unique taxa [19]. On this topic, Gweon and colleagues [17] stated that relatively low sequencing depths were sufficient to capture the broad-scale taxonomic composition of samples.

All pipelines used in this study demonstrated, on average, a low ability to recover pairs of species from the same genus, indicating that MAG recovery is not the best technology to study closely related species in microbial communities. Our results corroborate those of Sevim and collaborators [30], who suggested that genomes with higher sequence similarity lead to increased genome misassembly. However, their study did not consider the combination of species abundance, taxonomic relatedness, and sequencing depth. Nevertheless, communities following an exponential decay distribution with species abundance plateaus showed the highest ratio of species-pairs recovered irrespective of the pipeline. Further, the DT pipeline always yielded more TPs than the 8K and MM pipelines in any parameter combinations (species abundance versus taxonomy, sequencing depth versus species abundance distribution, equal abundance communities), which suggests that the combination of multiple binning tools is more accurate in characterizing community composition. Previous studies have suggested this increased accuracy due to the DT pipeline methodology [19,31].

In this study, we also aimed to determine how species not included in the simulated communities (False Positives) were recovered. None of our selected variables strongly influenced the number of FPs obtained by each pipeline, although the most significant differences were observed in communities sequenced at 60 million reads. However, the 8K pipeline always yielded more FPs than the DT and MM pipelines, independent of the community profile. Seven of the nine unique FP could be explained by misannotations since they belonged to the same genus as the original species. For example, *B. pertussis* is a Gram-negative bacterium that causes whooping cough [32] and belongs to the same genus as *B. bronchiseptica*, initially included in community A. In the case of *B. vaginale*, a Gram-positive bacterium that colonizes the vagina and contributes to the vaginal microbiome, studies have shown that bifidobacteria from vaginal and gut microbiomes are indistinguishable using comparative genomics [33]. The two remaining FPs shared the same family as the three original species. For example, *P. caldifontis* is a thermophilic Gram-negative bacterium commonly found in hot springs [34] and shares the same family as *P. thermarum* (included in the original communities). Because of these results, once important species are identified in the MAGs recovered, we suggest additional molecular tools such as qPCR or fluorescence in situ hybridization (FISH) should be employed to validate MAG recovery results [35,36].

We observed the influence of sequencing depth in communities following a logarithmic decay species abundance distribution. Bioreactors containing enriched microbial communities are clear examples of a logarithmic decay species abundance distribution since dominant species exist at sharply higher rates than others [37]. Thus, increasing sequencing depth will lower the detection limit, allowing the recovery of low-abundant species. Inversely, communities following an exponential decay species abundance distribution (e.g., communities not subject to conditions that favor only a few species) may comprise a higher number of species at similar abundances and a lesser fraction at low abundances. Trying to lower the detection limit by increasing sequencing depth will have little influence on the recovery of more species. The increase in sequencing depth may also increase the coverage and relative abundance of recovered species. However, the fact that some FPs present coverages and relative abundance greater than TPs by several folds suggests that robust community identification should not exclusively depend on these measures.

Of the 11 special cases (i.e., bacteria with linear or multiple chromosomes) added to the mock communities, only *Borrelia burgdorferi* and *Brucella ovis* were not recovered. The non-recovery of *Borrelia burgdorferi* may be linked to a misclassification rather than its linear

genome since we did recover *Borreliella burgdorferi* whose genome is also linear. The misclassification can also be tied to dividing the genus *Borrelia* into two genera, which has sparked some debate [38]. Similarly, the absence of *B. ovis* in our recovered MAGs may also be linked with misclassification due to high sequence similarities. Tsolis and colleagues [39] have shown that *B. ovis* shares similar chromosome sizes (bp), GC content, and number of protein-coding genes to other Brucella species such as *B. melitensis* (recovered in our data). Our data are insufficient to assess the influence of linear or multi-chromosome genomes on MAG recovery but suggest it may increase the challenge of taxonomic classification. However, current methods for MAG recovery do not address this issue. Thus, it would be beneficial for future studies to attempt to identify signatures that discriminate between linear and multiple chromosomes in MAGs.

While this study focused only on prokaryotes, similar studies of MAG recovery of Eukaryotes and viruses (DNA and RNA) may help improve ecological and biotechnological insights generated from genome-centric metagenomics.

Characterizing microbial communities is a challenging task due to their complexity. Our study evaluated the combined effect of species abundance, taxonomic proximity between species, and sequencing depth in prokaryotic MAG recovery. Our analyses suggested that none of the variables used (species abundance, taxonomic proximity, and sequencing depth) is solely responsible for MAG recovery or high accuracy in community characterization. However, our data suggested sequencing depth and taxonomic proximity between species influenced MAG recovery substantially more than species abundance. The point was supported by the very low percentages of taxonomically related "species pairs" recovered independently of the community profile or pipeline used. Integrating multiple binning tools via a consensus approach appears to perform better. Furthermore, our data indicated that coverage and relative abundance might not be the main drivers of MAG recovery since species with high abundances in the original communities did not necessarily guarantee their recovery. Additionally, the recovery of false positives with relative abundances and coverages many times greater than those of the original species suggests that species abundance is unreliable in characterizing a microbial community accurately. Interestingly, the difference in coverages for all "species pairs" recovered followed the same trend as their initial abundances in almost all communities, suggesting a slight connection between species abundance and genome coverage. Nevertheless, further studies need to be carried out to confirm this hypothesis. Genome linearity or multi-chromosomes had no visible effect on MAG recovery in our study. However, a more extensive study should be performed to fully determine if this is the case, including a larger number of genomes. In this study, we kept the number of original species constant. It would be important to include species number variations in future studies to assess ecological implications.

The pipeline used is also crucial since accuracy can range from 76% (when using the 8K pipeline) up to 92% (when using the DT pipeline). While the high accuracy obtained by the DT pipeline is promising, it is necessary to consider how the error rates become relevant for large MAG datasets. For every 10000 MAGs recovered, the number of false positive MAGs can range from 800 to 2400, depending on the pipeline. Therefore, our study demonstrated the need to verify findings based on MAGs, especially when novel species or special metabolic functions are highlighted.

Our study shows that approaches combining multiple binning tools perform best and should be favored in future studies. Additionally, a sequencing depth of 60 million reads appears to be a "sweet spot" for metagenome sequencing. Also, special care should be taken when performing MAG recovery on samples with relatively taxonomically homogeneous communities since taxonomic relatedness appears to have some influence on MAG recovery. ANI distances can be used in future studies, but researchers should be aware that these metrics have

yet to be employed for different taxonomic levels. Finally, special attention should be paid to making assumptions derived from MAG recovery. One can integrate rRNA gene reconstruction to strengthen the metagenomic study results.

## Methods

### Workflow

The workflow for this study is shown in Fig 5, and it was comprised of three main sections: pre-processing, binning, and post-processing.

### Pre-processing

**Selection of species of the in silico microbial communities.**   A total of 126 species were selected to represent the three main branches in the Tree of Life: 99 bacteria, 18 archaea, and nine fungi (S35 Table). Species were selected if a complete genome and the raw sequencing data were available in the National Center for Biotechnological Information (NCBI). Additionally, species were selected so that each community included taxonomic groups consisting of three species belonging to the same family and two of them to the same genus. On average, 10 MAGs are recovered from a single metagenomic library [40]. Thus, we divided these 126 species into three groups of 42 (33 bacteria, six archaea, and three fungi). A total of 12 taxonomic sets of species were present in each community. The three groups are intended to represent biological replicates. Next, each set of species was used to assemble communities with varying degrees of species abundance, taxonomic relatedness, and sequencing depth.

**Establishing taxonomic relatedness.**   Genomic similarity metrics such as Average Nucleotide Identity (ANI) can be used to differentiate species. However, to our knowledge, this has not been rigorously studied across different taxonomic levels. Thus, the taxonomic classification was used to create profiles of taxonomic relatedness. The species were grouped in triplets with different taxonomic distributions. First, a species was randomly set as the reference for a given phylum. Species belonging to the same genus as the reference were defined as "Very closely related", while species belonging to the same family (but not genus) as the reference were defined as "Closely related" (S30 Table). For example, the first triplet *Acetobacter aceti* would be randomly selected from the phylum Proteobacteria. The genome considered "Very closely related" would be *Acetobacter persici* since it belongs to the same genus. The genome considered "Closely related", would be *Gluconacetobacter diazotrophicus* since it belongs to the same family but not genus. The same procedure would be applied to all other genomes in each dataset. The communities were constructed according to five relationships between organisms: "Ordered", "Random", "Very closely related", "Closely related" and "Not closely related" (S36 Table). In the "Ordered" communities, species were simply placed based on their triplets. The "Random" communities were constructed without any specific criteria. The "Very closely related" communities were constructed considering their relationship with each other (e.g. species belonging to the same genus). Similarly, the "Closely related" communities were constructed considering the relationship between species at the family level. Lastly, "Not closely related" communities were constructed, so species with closely related relatives were not added at similar abundances. Examples of community definitions and relationships are shown in S36 Table. These community definitions and relationships did not include fungi and special cases of bacteria genome topology (consisting of linear or multiple chromosomes). However, representatives of species with linear (*Vibrio cholerae* and *Rhodobacter sphaeroides*) and multiple chromosomes (*Bacillus subtilis*, *Streptomyces griseus*, *Borrelia burgdorferi*, *Paracoccus denitrificans*, *Planktomarina temperate*, *Vibrio fluvialis*, and *Brucella melitensis*) were included in

**Selection of organisms**

**Community A**
33 bacteria
6 archaea
3 fungi

**Community B**
33 bacteria
6 archaea
3 fungi

**Community C**
33 bacteria
6 archaea
3 fungi

**Simulation of data**

Five Depths
180mi
120mi
60mi
30mi
10mi
Reads

Five Taxonomical Distributions (TD)
O  Ordered
C  Close Related
N  Not close Related
V  Very close Related
R  Randomly distributed

Five species abundance distributions (ED)
Equal abundance
Logarithmic decay
Exponential decay
Logarithmic decay w/ plateau
Exponential decay w/ plateau

69 simulated communities with different richness, evenness and taxonomical distribution

Quality Check: Removal of adapters and short reads

**Assembly of reads**

reads

scaffolds

**Genome recovery**

*DAS Tool (DT)\**

*Multi-metagenome (MM)\*\**

*8K pipeline (8K)\*\*\**

**Quality assessment - CheckM**

**Taxonomic classification - IDBA-UD**

**Fig 5. Workflow used in this study.** First, we proceeded with species selection and sequence retrieval from the National Center for Biotechnology Information (NCBI). Next, community profiles were generated based on species abundance, taxonomic distribution and sequencing depth. Metagenomes were simulated for each community profile using MetaSim [42]. A quality check was performed to remove adapters and short reads. The next step consisted of assembling reads into scaffolds and performing post-assembly quality checks. For genome recovery, three pipelines were used: DAS Tool (DT) [19], Multi-metagenome (MM) [20] and the pipeline used to recover more than 8000 metagenome-assembled genomes (MAGs) (8K) [12]. Completeness and contamination of MAGs was assessed using CheckM [44]. Taxonomic classification of the MAGs was performed by IDBA-UD [43].

each simulated community. The list of all species used per community and type of chromosome is available in S35 Table.

**Species abundance profiles.** The literature describes abundance and species phylogenies as the main parameters controlling MAG recovery [41]. In this study, we define community

species abundance as a function of abundance distribution and distribution arrays, i.e., which species occupy different levels of abundance in the community (Eq 1).

$$\text{Community species abundance} = f(\text{abundance distribution}, \text{ array distribution}) \qquad (1)$$

Species abundance refers to the relative number of organisms of each species inside a given environment (system), which, in this case, is the main degree of freedom that will vary amongst the different communities. In a metagenomic community, species abundance and abundance can sometimes be used interchangeably. The abundance of genomic data can be evenly distributed (equal species abundance), normally or disruptively. In this study, we simulated abundance in several different states of species abundance distribution (equal species abundance, logarithmic decay, exponential decay, logarithmic decay with abundance plateaus, and exponential decay with abundance plateaus). Communities classified as logarithmic decay with abundance plateaus are composed of species whose abundance follows a logarithmic decrease but with intervals of equal abundances for pairs of species. Similarly, communities classified as exponential decay with abundance plateaus are composed of species whose abundance followed an exponential decrease but with intervals of equal abundances for pairs of species (S21 Fig). Additionally, we coupled different abundance profiles with different taxonomic distribution profiles. The exact abundances used for each species for each generated community profile is in S37 Table.

**Sequencing depth.**    To evaluate the influence of sequencing on the recovery of MAGs, we simulated communities with five sequencing depths (10, 30, 60, 120, and 180 million paired-end ILLUMINA reads, 2x150bp).

**In silico metagenomic library preparation.**    We generated 23 libraries for each group of species (hereafter, A, B, and C): 12 through the combination of four species abundance distribution profiles (logarithmic decay, exponential decay, logarithmic decay with abundance plateaus, and exponential decay with abundance plateaus) with three taxonomic relatedness profiles ("Very closely related ", "Closely related" and "Not closely related") at a depth of 60 million reads. Ten libraries were generated combining the logarithmic decay and exponential decay species abundance profiles with all the possible sequencing depths (10, 30, 60, 120, and 180 million reads) with no specific criteria for the species taxonomic relatedness ("Random" taxonomic distribution profile). A single library was generated with all species at equal abundances, "Ordered" taxonomy, and a sequencing depth of 60 million reads. This library was used to test MAG recovery at identical species abundances.

## Processing

**Simulation of High-throughput sequencing data.**    The reads were generated using the MetaSIM Sequencing simulator (v0.9.1) [42]. The fragment size was set to 180 bp with a read size of 125 bp.

**Assembly.**    The reads were assembled into scaffolds of each simulated metagenomic library using IDBA-UD [43] with default parameters.

**Draft genome reconstruction pipeline.**    We used three pipelines to recover high-quality metagenome-assembled genomes (MAGs). The multi-metagenome (MM) pipeline [20] assembles near-complete draft genomes in a two-step process (i.e., primary binning is performed independently of sequence composition followed by refinement of population genomes using sequence composition-dependent methods and visualization tools). The DAS Tool (DT) pipeline [19] generates bins by integrating multiple binning algorithms. The pipeline developed by Parks and co-workers (8K) [12] uses MetaBAT [13] and integrates genome abundances and tetranucleotide frequencies to generate bins.

**Bin quality assessment and filtering.**    Completeness and contamination measures were obtained via CheckM [44]. Bin quality was determined using the approach by Parks and colleagues [12] (Eq 2). Only bins with a quality score above 50 were considered for subsequent analyses.

$$\text{Quality score} = \text{Completeness} - (5 \times \text{Contamination}) \tag{2}$$

**Assessing the performance of binning pipelines.**    We classified each genome used in the different communities with GTDB-tk version 1.3 [45] to determine the performance of each pipeline and the influence of species abundances, taxonomic relatedness, and sequencing depth. True positives (TP) were defined as MAGs correctly classified in each community, false positives (FP) were defined as MAGs with taxonomy not included in the original communities, and false negatives (FN) were defined as MAGs present in the original community but not found after the recovery process. Additionally, to test the effect of closely and very closely related species at equal or similar abundances on genome recovery, we calculated the fraction of the pairs of species that were recovered in the same community. We determined the accuracy of MAG recovery using Eq 3:

$$\text{Accuracy (\%)} = (\text{TP}/(\text{TP} + \text{FP})) \times 100 \tag{3}$$

**Detection limits.**    Usually, the number of species recovered in a sample using metagenomics is a product of the number of reads sequenced [46]. The higher the number of reads, the more likely we can recover species at low quantities. To test this hypothesis, we calculated each MAG's coverage (Eq 4) and relative abundance (Eq 5) in their respective library.

$$\text{Coverage} = \text{L} \times \text{N}/\text{G} \tag{4}$$

In Eq 4, L is the mean length of the reads per library, N is the total number of mapped reads in the MAG, and G is the size of the MAG (in base pairs).

$$\text{Relative abundance} = \text{N}/\text{R} \tag{5}$$

In Eq 5, N is the total number of mapped reads in the MAG, and R is the total number of reads in a sample.

**Statistical analysis.**    We used Student's bonferroni adjusted t-test to determine significant differences in the number of MAGs, TPs, and FPs when performing pairwise comparisons of sequencing depth, taxonomic relatedness, and species abundance distribution profiles. Results were considered statistically significant when the adjusted P values were below 0.05. Additionally, we performed ANOVA followed by Tukey's test for the pairwise comparisons. The results with both methods and scripts used to perform the analyses are available at https://github.com/mdsufz/mockc_analysis

**16S rRNA reconstruction.**    16S rRNA sequences were recovered using the ssu_finder function from CheckM [44].

**Computational resources.**    All *in silico* operations were performed using a High-Performance Computer (HPC). The HPC cluster was composed of 44 nodes with 28 cores each. Each node had 224 GB of main memory usable for jobs.

## Supporting information

**S1 Fig.** Completeness and contamination percentages of all Metagenome-Assembled Genomes (MAGs) recovered by each binning pipeline (8K DT and MM) (A). The quality

MAGs are also color-coded for their respective quality score (B) (Quality score = completeness —5 * contamination). The normalized counts of MAGs are shown according to completeness (C) and contamination (D).
(TIF)

**S2 Fig. Sankey plot showing taxonomic classification of Metagenome-Assembled Genomes (MAGs) generated by the pipeline Parks and colleagues (8K) from species group A.**
(TIF)

**S3 Fig. Sankey plot showing taxonomic classification of Metagenome-Assembled Genomes (MAGs) generated by the pipeline Parks and colleagues (8K) from species group B.**
(TIF)

**S4 Fig. Sankey plot showing taxonomic classification of Metagenome-Assembled Genomes (MAGs) generated by the pipeline Parks and colleagues (8K) from species group C.**
(TIF)

**S5 Fig. Sankey plot showing taxonomic classification of Metagenome-Assembled Genomes (MAGs) generated by the pipeline Sieber and colleagues (DT) from species group A.**
(TIF)

**S6 Fig. Sankey plot showing taxonomic classification of Metagenome-Assembled Genomes (MAGs) generated by the pipeline Sieber and colleagues (DT) from species group B.**
(TIF)

**S7 Fig. Sankey plot showing taxonomic classification of Metagenome-Assembled Genomes (MAGs) generated by the pipeline Sieber and colleagues (DT) from species group C.**
(TIF)

**S8 Fig. Sankey plot showing taxonomic classification of Metagenome-Assembled Genomes (MAGs) generated by the pipeline Albertsen and colleagues (MM) from species group A.**
(TIF)

**S9 Fig. Sankey plot showing taxonomic classification of Metagenome-Assembled Genomes (MAGs) generated by the pipeline Albertsen and colleagues (MM)from species group B.**
(TIF)

**S10 Fig. Sankey plot showing taxonomic classification of Metagenome-Assembled Genomes (MAGs) generated by the pipeline Albertsen and colleagues (MM)from species group C.**
(TIF)

**S11 Fig. Student's Bonferroni adjusted t-test comparing metagenome-assemble genome counts obtained by the MM pipeline in communities sequenced at different sequencing depths (10 30 60 120 and 180 million reads) following Logarithmic decay and exponential decay species abundance distributions.** Taxonomic distribution was set to Random. (*P-value < 0.05)
(TIF)

**S12 Fig. Student's Bonferroni adjusted t-test comparing metagenome-assemble genome counts obtained by the MM pipeline in communities following different species abundance distributions (Logarithmic decay; Exponential decay; Logarithmic decay with abundance plateaus; Exponential decay with abundance plateaus) for each taxonomic distribution (Not related; Closely related; Very closely related).** Sequencing depth of the communities

was kept at 60 million reads. (*P-value < 0.05)
(TIF)

**S13 Fig. Student's Bonferroni adjusted t-test comparing metagenome-assemble genome counts obtained by the DT pipeline in communities following different species abundance distributions (Logarithmic decay; Exponential decay) and sequenced at different depths (10 30 60 120 and 180 million reads).** Species taxonomy was set to Random. (* P-value < 0.05)
(TIF)

**S14 Fig. Student's Bonferroni adjusted t-test comparing metagenome-assemble genome counts obtained by the DT pipeline in communities following different species abundance distributions (Logarithmic decay; Exponential decay; Logarithmic decay with abundance plateaus; Exponential decay with abundance plateaus) for each taxonomic distribution (Not related; Closely related; Very closely related).** Sequencing depth of the communities was kept at 60 million reads. (* P-value < 0.05)
(TIF)

**S15 Fig. Student's Bonferroni adjusted t-test comparing True Positives (Taxonomies recovered and in the original communities) obtained by the DT pipeline in communities sequenced at different sequencing depths (10 30 60 120 and 180 million reads) following Logarithmic decay (2) and exponential decay species abundance distributions.** (*P-value < 0.05)
(TIF)

**S16 Fig. Student's Bonferroni adjusted t-test comparing True Positives (Taxonomies recovered and in the original communities) obtained by the MM pipeline in communities following different species abundance distributions (Logarithmic decay; Exponential decay; Logarithmic decay with abundance plateaus; Exponential decay with abundance plateaus) for each taxonomic distribution (Not related; Closely related; Very closely related).** Sequencing depth of the communities was kept at 60 million reads. (*P-value < 0.05)
(TIF)

**S17 Fig. Student's Bonferroni adjusted t-test comparing True Positives (Taxonomies recovered and in the original communities) obtained by the MM pipeline in communities sequenced at different sequencing depths (10 30 60 120 and 180 million reads) following Logarithmic decay and exponential decay species abundance distributions.** (*P-value < 0.05)
(TIF)

**S18 Fig. Student's Bonferroni adjusted t-test comparing True Positives (Taxonomies recovered and in the original communities) between all pipelines used (8K DT and MM) in communities sequenced at different sequencing depths (10 30 60 120 and 180 million reads) following Logarithmic decay and exponential decay species abundance distributions.** (*P-value < 0.05)
(TIF)

**S19 Fig. Student's Bonferroni adjusted t-test comparing False Positives (Taxonomies recovered but not in the original communities) between all pipelines used (8K DT and MM) in communities sequenced at different sequencing depths (10 30 60 120 and 180 million reads) following logarithmic decay and exponential decay species abundance**

**distributions.** (*P-value < 0.05)
(TIF)

**S20 Fig. Student's Bonferroni adjusted t-test comparing the recovery of "pairs" of species between all pipelines used (8K DT and MM) according to Taxonomic distribution (Closely related; Very closely related) and species abundance distribution (Logarithmic decay; Exponential decay; Logarithmic decay with abundance plateaus; Exponential decay with abundance plateaus).** Sequencing depth of the communities was kept at 60 million reads. (*P-value < 0.05)
(TIF)

**S21 Fig. Species abundance distribution profiles used in this study.** Abundance distribution 1: Equal abundance (all species are equally abundant); Species abundance distribution 2: Logarithmic decay; Species abundance distribution 3: Exponential decay; Species abundance distribution 4: Logarithmic decay with plateaus (Some species have equal abundances); Species abundance distribution 5: Exponential decay with plateaus (Some species have equal abundances).
(TIF)

**S22 Fig. Student's Bonferroni adjusted t-test comparing True Positives (Species recovered and in the original communities) between all pipelines used (8K DT and MM) in communities composed of species with equal abundance (i.e. equal species abundance distribution).** The sequencing depth of the communities was kept at 60 million reads. (*P-value < 0.05).
(TIF)

**S1 Table. Metagenome-assembled genomes (MAGs) quality assessment (Completeness, contamination and strain heterogeneity) obtained for each community profile.** Features include species group (Community), Species abundance distribution, Taxonomic distribution, Sequencing depth, Pipeline and genomic metrics (number of contigs, N50, L50, average and standard deviation of GC content))
(XLSX)

**S2 Table. Values of the total number of Metagenome-Assembled Genomes (MAGs) recovered by each pipeline (8K, DT and MM), the number of original species found (True Positives), the number of species obtained not present in the original community (False positives), the number of original species not recovered, average number of MAGs per species and ratio of FP according to each community profile (Communities A, B and C; Sequencing depth; Taxonomic relatedness (Closely related, Very closely related, Not closely related, Ordered, Randomly assigned) and species abundance distribution (Equal abundance, Logarithmic decay, Exponential decay, Logarithmic decay with abundance plateaus, Exponential decay with abundance plateaus).** Sequencing depth was kept at 60 million reads. (Ratio FP = total number of FPs/total number of species recovered (*100); Accuracy = (TP/TP+FP)*100)
(XLSX)

**S3 Table. Taxonomic classification of 16S rRNA sequences recovered from each MAG and confidence score and the respective taxonomic classification of Metagenome-Assembled Genomes (MAGs) in each community (A,B and C) using each tool (8K, DT and MM).** MAG ID contemplates the community (A, B, C), species abundance distribution (Equal abundance; Logarithmic decay; Exponential decay; Logarithmic decay with abundance plateaus; Exponential decay with abundance plateaus), Taxonomic relatedness (Ordered; Not related;

Closely related; Very closely related; Random) and sequencing depth (10, 30, 60, 120 and 1280 million reads). 16S rRNa sequences were recovered using the ssu_finder function from CheckM. Classification of 16S rRNA sequences was determined using the online version of SILVA (https://www.arb-silva.de/aligner/).
(XLSX)

**S4 Table. Taxonomic classification of Metagenome-Assembled Genomes (MAGs) in each community (A,B and C) using each tool (8K, DT and MM).** Features include Species abundance distribution (Equal abundance; Logarithmic decay; Exponential decay; Logarithmic decay with abundance plateaus; Exponential decay with abundance plateaus), Taxonomic distribution (Ordered; Not related; Closely related; Very closely related; Random), quality metrics (completeness, contamination and strain heterogeneity) and abundance metrics (coverage and relative abundance).
(XLSX)

**S5 Table. Student's Bonferroni adjusted t-test comparing metagenome-assemble genome counts obtained by the 8K pipeline in communities following different species abundance distributions (Logarithmic decay; Exponential decay; Logarithmic decay with abundance plateaus; Exponential decay with abundance plateaus) for each taxonomic distribution (Not related; Closely related; Very closely related).** Sequencing depth of the communities was kept at 60 million reads.
(XLSX)

**S6 Table. Student's Bonferroni adjusted t-test comparing metagenome-assemble genome counts obtained by the 8K pipeline in communities composed of species with different taxonomic proximity levels (Not related; Closely related; Very closely related) following different species abundance distributions (Equal evenness; Logarithmic decay; Exponential decay; Logarithmic decay with abundance plateaus; Exponential decay with abundance plateaus).** Sequencing depth of the communities was kept at 60 million reads.
(XLSX)

**S7 Table. Student's Bonferroni adjusted t-test comparing metagenome-assemble genome counts obtained by the 8K pipeline in communities sequenced at different sequencing depths (10, 30, 60, 120 and 180 million reads) following Logarithmic decay and exponential decay species abundance distributions.**
(XLSX)

**S8 Table. Student's Bonferroni adjusted t-test comparing metagenome-assemble genome counts obtained by the MM pipeline in communities sequenced at different sequencing depths (10, 30, 60, 120 and 180 million reads) following Logarithmic decay and exponential decay species abundance distributions.**
(XLSX)

**S9 Table. Student's Bonferroni adjusted t-test comparing metagenome-assemble genome counts obtained by the MM pipeline in communities following different species abundance distributions (Logarithmic decay; Exponential decay; Logarithmic decay with abundance plateaus; Exponential decay with abundance plateaus) for each taxonomic distribution (Not related; Closely related; Very closely related).** Sequencing depth of the communities was kept at 60 million reads.
(XLSX)

**S10 Table. Student's Bonferroni adjusted t-test comparing metagenome-assemble genome counts obtained by the DT pipeline in communities sequenced at different sequencing depths (10, 30, 60, 120 and 180 million reads) following Logarithmic decay and exponential decay species abundance distributions.**
(XLSX)

**S11 Table. Student's Bonferroni adjusted t-test comparing metagenome-assemble genome counts obtained by the DT pipeline in communities following different species abundance distributions (Logarithmic decay; Exponential decay; Logarithmic decay with abundance plateaus; Exponential decay with abundance plateaus) for each taxonomic distribution (Not related; Closely related; Very closely related).** Sequencing depth of the communities was kept at 60 million reads.
(XLSX)

**S12 Table. Student's Bonferroni adjusted t-test comparing metagenome-assemble genome counts between all pipelines used (8K, DT and MM) in communities composed of species with equal abundance (i.e., equal species abundance distribution).** Sequencing depth of the communities was kept at 60 million reads.
(XLSX)

**S13 Table. Student's Bonferroni adjusted t-test comparing metagenome-assemble genome counts between all pipelines used (8K, DT and MM) according to species abundance distribution (Logarithmic decay; Exponential decay) and sequencing depth (10, 30, 60, 120 and 180 million reads).** Taxonomic relatedness was kept random.
(XLSX)

**S14 Table. Student's Bonferroni adjusted t-test comparing metagenome-assemble genome counts between all pipelines used (8K, DT and MM) according to Taxonomic distribution (Not related; Closely related; Very closely related) and species abundance distribution (Logarithmic decay; Exponential decay; Logarithmic decay with abundance plateaus; Exponential decay with abundance plateaus).** Sequencing depth of the communities was kept at 60 million reads.
(XLSX)

**S15 Table. Student's Bonferroni adjusted t-test comparing True Positives (Species recovered and in the original communities) obtained by the 8K pipeline in communities following different species abundance distributions (Logarithmic decay; Exponential decay; Logarithmic decay with abundance plateaus; Exponential decay with abundance plateaus) for each taxonomic distribution (Not related; Closely related; Very closely related).** Sequencing depth of the communities was kept at 60 million reads.
(XLSX)

**S16 Table. Student's Bonferroni adjusted t-test comparing True Positives (Species recovered and in the original communities) obtained by the 8K pipeline in communities sequenced at different sequencing depths (10, 30, 60, 120 and 180 million reads) following Logarithmic decay and exponential decay species abundance distributions.**
(XLSX)

**S17 Table. Student's Bonferroni adjusted t-test comparing True Positives (Species recovered and in the original communities) obtained by the 8K pipeline in communities composed of species with different taxonomic proximity levels (Not related; Closely related; Very closely related) following different species abundance distributions (Logarithmic**

decay; Exponential decay; Logarithmic decay with abundance plateaus; Exponential decay with abundance plateaus). Sequencing depth of the communities was kept at 60 million reads.
(XLSX)

**S18 Table. Average values for MAGs recovered, original species found and number of bins wrongly classified, per community profile (Sequencing depth, taxonomic distribution and species abundance distribution), per pipeline.** Additionally, standard deviation errors are also presented for each category.
(XLSX)

**S19 Table. Student's Bonferroni adjusted t-test comparing True Positives (Species recovered and in the original communities) obtained by the DT pipeline in communities following different species abundance distributions (Logarithmic decay; Exponential decay; Logarithmic decay with abundance plateaus; Exponential decay with abundance plateaus) for each taxonomic distribution (Not related; Closely related; Very closely related).** Sequencing depth of the communities was kept at 60 million reads.
(XLSX)

**S20 Table. Student's Bonferroni adjusted t-test comparing True Positives (Species recovered and in the original communities) obtained by the DT pipeline in communities sequenced at different sequencing depths (10, 30, 60, 120 and 180 million reads) following Logarithmic decay and exponential decay species abundance distributions.**
(XLSX)

**S21 Table. Student's Bonferroni adjusted t-test comparing True Positives (Species recovered and in the original communities) obtained by the MM pipeline in communities following different species abundance distributions (Logarithmic decay; Exponential decay; Logarithmic decay with abundance plateaus; Exponential decay with abundance plateaus) for each taxonomic distribution (Not related; Closely related; Very closely related).** Sequencing depth of the communities was kept at 60 million reads.
(XLSX)

**S22 Table. Student's Bonferroni adjusted t-test comparing True Positives (Species recovered and in the original communities) obtained by the MM pipeline in communities sequenced at different sequencing depths (10, 30, 60, 120 and 180 million reads) following Logarithmic decay and exponential decay species abundance distributions.**
(XLSX)

**S23 Table. Student's Bonferroni adjusted t-test comparing True Positives (Species recovered and in the original communities) obtained by the MM pipeline in communities composed of species with different taxonomic proximity levels (Not related; Closely related; Very closely related) following different species abundance distributions (Logarithmic decay; Exponential decay; Logarithmic decay with abundance plateaus; Exponential decay with abundance plateaus).** Sequencing depth of the communities was kept at 60 million reads.
(XLSX)

**S24 Table. Student's Bonferroni adjusted t-test comparing True Positives (Species recovered and in the original communities) between all pipelines used (8K, DT and MM) in communities composed of species with equal abundance (i.e., equal abundance**

**distribution).** Sequencing depth of the communities was kept at 60 million reads.
(XLSX)

**S25 Table. Student's Bonferroni adjusted t-test comparing True Positives (Species recovered and in the original communities) between all pipelines used (8K, DT and MM) according to Taxonomic distribution (Not related; Closely related; Very closely related) and species abundance distribution (Logarithmic decay; Exponential decay; Logarithmic decay with abundance plateaus; Exponential decay with abundance plateaus).** Sequencing depth of the communities was kept at 60 million reads.
(XLSX)

**S26 Table. Student's Bonferroni adjusted t-test comparing True Positives (Species recovered and in the original communities) between all pipelines used (8K, DT and MM) in communities sequenced at different sequencing depths (10, 30, 60, 120 and 180 million reads) following Logarithmic decay and exponential decay species abundance distributions.**
(XLSX)

**S27 Table. Student's Bonferroni adjusted t-test comparing False Positives (Species recovered but not in the original communities) between all pipelines used (8K, DT and MM) in communities sequenced at different sequencing depths (10, 30, 60, 120 and 180 million reads) following Logarithmic decay and exponential decay species abundance distributions.**
(XLSX)

**S28 Table. Species populating the original communities recovered by each pipeline.** (* Special cases: linear chromosomes or multiple chromosomes; **: Eukaryotes; ü: recovered; x: not recovered,NA: not applicable (i.e., not in the original community)).
(XLSX)

**S29 Table. Percentage of recovery of taxonomically related species (Very closely related; Closely related) by each pipeline (8K, DT and MM) using different species abundance distribution profiles (Logarithmic decay; Exponential decay; Logarithmic decay with abundance plateaus; Exponential decay with abundance plateaus).** Sequencing depth was kept at 60 million end reads.
(XLSX)

**S30 Table. Triplets of species used in each community and their taxonomic relatedness.** The "Reference" column is populated with the species to which the species in "Very Closely Related" and "Closely Related" are related at the Genus and Family level, respectively. Additional features include the Domain, Phylum and community set (A,B and C) in which those species were used.
(XLSX)

**S31 Table. Recovery of pairs of genomes by the 8K pipeline according to their taxonomic distribution (Closely related—Reference and paired genome share the same genus; Very closely related—Reference and paired genome share the same family).** Additional features include the coverage and relative abundance of the species in the original communities. Community species dataset: Original species list. Species abundance distribution (Logarithmic decay; Exponential decay; Logarithmic decay with abundance plateaus; Exponential decay with abundance plateaus). Sequencing depth was kept at 60 million reads.
(XLSX)

**S32 Table. Recovery of pairs of genomes by the DT pipeline according to their taxonomic distribution (Closely related—Reference and paired genome share the same genus; Very closely related—Reference and paired genome share the same family).** Additional features include the coverage and relative abundance of the species in the original communities. Community species dataset: Original species list. Species abundance distribution (Logarithmic decay; Exponential decay; Logarithmic decay with abundance plateaus; Exponential decay with abundance plateaus). Sequencing depth was kept at 60 million reads.
(XLSX)

**S33 Table. Recovery of pairs of genomes by the MM pipeline according to their taxonomic distribution (Closely related—Reference and paired genome share the same genus; Very closely related—Reference and paired genome share the same family).** Additional features include the coverage and relative abundance of the species in the original communities. Community species dataset: Original species list. Species abundance distribution (Logarithmic decay; Exponential decay; Logarithmic decay with abundance plateaus; Exponential decay with abundance plateaus). Sequencing depth was kept at 60 million reads.
(XLSX)

**S34 Table. Student's Bonferroni adjusted t-test comparing the recovery of "pairs" of Closely related and Very closely related species between all pipelines used (8K, DT and MM) and species abundance distribution (Logarithmic decay; Exponential decay; Logarithmic decay with abundance plateaus; Exponential decay with abundance plateaus).** Sequencing depth of the communities was kept at 60 million reads.
(XLSX)

**S35 Table. Domain, name, chromosome type, assembly identifier of all species used in this study.** Further, the community set in which the species were placed is also shown.
(XLSX)

**S36 Table. Community definitions and relationships.** Very closely related: Communities composed of species belonging to the same genus as the reference; Closely related: Communities composed of species belonging to the same family (but not genus) as the reference; Not closely related: Communities not satisfying none of the genus and family conditions in Very closely related and Closely related; Randomly assigned: Communities assembled with randomly organized species; Ordered: no criteria for distribution but the community order itself (only used on equal evenness communities). Letters represent the triplets used (.e.g, A1 (reference genome), A2 (species with the same genus as A1), and A3 (species with the same family but not genus of A1)).
(XLSX)

**S37 Table. Relative abundance from the originally generated community profile (Communities A, B and C; Sequencing depth; Taxonomic relatedness (Closely related, Very closely related, Not closely related, Ordered, Randomly assigned) and species abundance distribution (Equal abundance, Logarithmic decay, Exponential decay, Logarithmic decay with abundance plateaus, Exponential decay with abundance plateaus).**
(XLSX)

## Author Contributions

**Conceptualization:** Ulisses Rocha, Joao Pedro Saraiva.

**Data curation:** Rodolfo Toscan.

**Formal analysis:** Jonas Coelho Kasmanas, Joao Pedro Saraiva.

**Funding acquisition:** Ulisses Rocha.

**Project administration:** Ulisses Rocha.

**Resources:** Rodolfo Toscan.

**Supervision:** Ulisses Rocha, Joao Pedro Saraiva.

**Validation:** Jonas Coelho Kasmanas.

**Writing – original draft:** Ulisses Rocha, Joao Pedro Saraiva.

**Writing – review & editing:** Ulisses Rocha, Danilo S. Sanches, Stefania Magnusdottir.

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
