## [Decision Letter · Decision Letter 0]

2 Oct 2023

Dear Dr. Saraiva,

Thank you very much for submitting your manuscript "Simulation of 69 microbial communities indicates sequencing depth and false positives are major drivers of bias in Prokaryotic metagenome-assembled genome recovery" for consideration at PLOS Computational Biology.

As with all papers reviewed by the journal, your manuscript was reviewed by members of the editorial board and by several independent reviewers. In light of the reviews (below this email), we would like to invite the resubmission of a significantly-revised version that takes into account the reviewers' comments.

Please note that while there is enthusiasm from all three reviewers about the focus of this work, there are substantial critiques on methodology and presentation.  So, while this is a "major revision" decision, please make note of the expectation for extensive revision for the manuscript to be considered further.  Please also note the repeated request for the code to be available per the journal policy.

We cannot make any decision about publication until we have seen the revised manuscript and your response to the reviewers' comments. Your revised manuscript is also likely to be sent to reviewers for further evaluation.

Sincerely,

Jason Papin

Editor-in-Chief

PLOS Computational Biology

Reviewer's Responses to Questions

**Comments to the Authors:**

Reviewer #1: See attached document.

Reviewer #2: The generation of Metagenome-Assembled Genomes (MAGs) from individual microbes within complex samples is a critical step for identifying the community members present and their functional capacity within a metagenomics sample. Previous work has shown that intrinsic parameters of the sample or the sequencing can bias MAG generation, which would significantly alter the interpretation of a metagenomics experiment’s results. The manuscript by da Rocha et al aims to investigate bias in tools used for MAG generation. The authors generated in silico microbial communities, allowing them to test three MAG generation pipelines for the effect of sample evenness, sequencing depth, and taxonomic relatedness on MAG recovery. This work is of significant interest to individuals working with metagenomics data, as it tests the influence of these factors on bias in MAG recovery as well as tests the accuracy of several MAG generation pipelines on simulated data.

Overall, the authors found that the DAS Tool (DT) pipeline had the highest accuracy of MAG generation pipelines tested. The authors also found other associations between the number of MAGs recovered and other parameters, although there were few consistent trends. Of note, the statistical approach used is not suited for the number of comparisons made in this analysis, making it difficult to interpret these results. In addition, the language of the manuscript could be revised to ease interpretation of the results. Comments are listed below.

Major Comments

1. Throughout the paper, the authors use a Student’s t test to test for significant differences between groups. However, a Student’s t test is only valid for comparisons between two groups, and most of these comparisons involve three or more comparisons. Therefore, the statistics should be revised for this study to use an alternative test, such as a One-way ANOVA, instead.

2. While the authors draw many conclusions on the associations between sample evenness, sequencing depth, and taxonomic similarity, most of this data is presented in supplementary figures, with only one figure (Fig. 3) showing significant differences across these parameters. I think some of this data should be presented within the body of the manuscript as main figures to ease interpretation of the work.

3. The text should be revised to more clearly explain the main takeaways of the paper. For example, more detailed explanation of the evenness models investigated should be included within the results section.

Minor Comments

1. On line 35, the final sentence of the abstract is incomplete and should be revised.

2. The MAGs generated for this manuscript have been shared publicly on NCBI Bioproject, but no code for this analysis was listed. Are the tools used for this analysis all stand-alone applications, and thus no custom code was required?

Reviewer #3: In this work, de Rocha et al. test three different pipelines for extracting metagenome-assembled genomes (MAGs) from three metagenomic datasets while systematically varying the input parameters for the pipelines. Their aim was to evaluate how sample evenness, sequencing depth, and taxonomic distribution profiles influence the recovery of genomes. To accomplish this, they generated mock microbial communities by dividing 126 bacteria, archaea, and fungi into three groups of 42 each. They then tested three pipelines, 8K, DT, and MM, with sequencing depth, taxonomical distribution, and evenness distribution varied with five different parameters each. The results indicated that the 8Kpipeline recovered the highest number of MAGs but only 25% of the 42 species on average, while the DT pipeline recovered 45% of species on average. The 8K pipeline also recovered the lowest number of true positives and had the highest number of false negatives. Moreover, MAG recovery was was influenced by taxonomic relatedness and sequencing depth but not evenness distribution. Overall, their results suggested that the DT pipeline has the highest accuracy and sequencing depth influences results. The three pipelines also had generally low accuracy in recovering closely related species.

This is a timely study that demonstrates the importance of the choice of reconstruction pipeline and parameter selection. As MAG reconstruction is commonly used by the research community, this is valuable knowledge.

Main comment:

- While the MAGs have been made available, the code used to generate and evaluate the MAGs does not appear to be available. This is also important for the audience to reproduce how the quality and accuracy of the MAGs was determined.

Minor comments:

- Line 36: incomplete sentence “Our data indicates the scientific community should their findings from MAG recovery”

- There are a few typos (e.g., “Informtaion” in line 50).

- Line 104: other than in the abstract, this is the first time, the abbreviations for the three pipelines are used before being defined, which is a bit confusing.

- It would be helpful to have the overview on the workflow near the beginning of the manuscript rather than as Figure 6.

- There is a high number of supplementary tables most of which are small. I would suggest condensing them.

**Have the authors made all data and (if applicable) computational code underlying the findings in their manuscript fully available?**

Reviewer #1: **No: **Data to produce the plots are provided, as well as the sequencing data. However, there is not a code availability statement and no link to the code for bioinformatic workflow, statistical analysis and plot generation.

Reviewer #2: Yes

Reviewer #3: **No: **While the MAGs have been shared, the code for performing the simulations does not seem to be available.

PLOS authors have the option to publish the peer review history of their article (what does this mean?). If published, this will include your full peer review and any attached files.

Reviewer #1: No

Reviewer #2: No

Reviewer #3: No
---

## [Decision Letter · Decision Letter 1]

13 Aug 2024

Dear Dr Rocha,

Thank you very much for submitting your manuscript "Simulation of 69 microbial communities indicates sequencing depth and false positives are major drivers of bias in Prokaryotic metagenome-assembled genome recovery" for consideration at PLOS Computational Biology. As with all papers reviewed by the journal, your manuscript was reviewed by members of the editorial board and by several independent reviewers. The reviewers appreciated the attention to an important topic but raised important points that need to be further addressed.

Sincerely,

Jason Papin

Editor-in-Chief

PLOS Computational Biology

Reviewer's Responses to Questions

**Comments to the Authors:**

Reviewer #2: This manuscript evaluates several Metagenome-Assembled Genome pipelines and the effects of sequence depth, species abundance, and species relatedness on model performance. In this revision, the authors put substantial effort into revising the organization of the manuscript, and I found that the goals of the manuscript and major findings were much clearer and easier to follow. The authors also added a new Figure 1, which highlights the major differences across pipelines. However, the statistical approach used in the manuscript is still a concern, as it underestimates the possibility of Type 1 errors in the results. Comments are listed below:

Major Comments

1. A Student’s t test is used for nearly all of the analyses in the manuscript, despite most of these analyses comparing differences between 3 or more groups (the paper compared outcomes across 3 different MAG pipelines, 5 different sequence depths, and 4 different abundance models). While I understand that the pairwise comparisons between two groups is what you are interested in, using a t test across all comparisons will overestimate the significance of your results. This is because each comparison is associated with a Type 1 error, where you falsely reject the null hypothesis. The probability of a Type 1 error will increase with each comparison, and when you are performing 3 tests (i.e., 8K vs DT, 8K vs MM, DT vs MM) with the same data, using a p value threshold of p < 0.05 will underestimate this error rate. The reported p values are also not corrected for multiple comparisons, which occur when using a univariate test repeatedly across many comparisons. As is, the p values presented overestimate the number of statistical differences and must be adjusted to account for these issues.

A One-way ANOVA with a post hoc test (such as Tukey’s method) will adjust the p value to account for the increased Type 1 error rate, and pairwise comparisons can be performed to identify statistical differences between paired conditions. For example, in Supplemental Table 6, a One-way ANOVA should be used to calculate pairwise comparisons across all Taxonomic Distributions for each Species Abundance. I would recommend this approach to account for the increased Type 1 error rate.

Minor Comments

1. The resolution of the figures (particularly Figure 1 and Figure 4) appears low. Please check figure resolution before submission of final figures.

Reviewer #3: The authors have done a good job revising the manuscript and addressing my and the other reviewers' comments. I have no further comments.

**Have the authors made all data and (if applicable) computational code underlying the findings in their manuscript fully available?**

Reviewer #2: Yes

Reviewer #3: Yes

PLOS authors have the option to publish the peer review history of their article (what does this mean?). If published, this will include your full peer review and any attached files.

Reviewer #2: No

Reviewer #3: No

Figure Files:

Data Requirements:

Reproducibility:

References:

---

## [Editor Report · Decision Letter 2]

1 Oct 2024

Dear Dr Rocha,

We are pleased to inform you that your manuscript 'Simulation of 69 microbial communities indicates sequencing depth and false positives are major drivers of bias in Prokaryotic metagenome-assembled genome recovery' has been provisionally accepted for publication in PLOS Computational Biology.

Best regards,

Jason Papin

Editor-in-Chief

PLOS Computational Biology

---

## [Editor Report · Acceptance letter]

17 Oct 2024

PCOMPBIOL-D-23-00832R2 

Simulation of 69 microbial communities indicates sequencing depth and false positives are major drivers of bias in Prokaryotic metagenome-assembled genome recovery

Dear Dr Rocha,

I am pleased to inform you that your manuscript has been formally accepted for publication in PLOS Computational Biology. Your manuscript is now with our production department and you will be notified of the publication date in due course.

With kind regards,

Anita Estes
